# How muscle ageing affects rapid goal-directed movement: mechanistic insights from a simple model

Delyle T. Polet[¤]*, Christopher T. Richards[¤]

Structure and Motion Laboratory, Department of Comparative Biomedical Sciences, Royal Veterinary College, London, United Kingdom

¤ Current address: Department of Bioengineering, Imperial College London, London, United Kingdom
* delylepolet@gmail.com

## Abstract

As humans and other animals age, passive and active muscle properties change markedly, with reduced peak tension, peak strain rate, activation and deactivation rate, and increased parallel stiffness. It is thought that these alterations modify locomotor performance, but establishing causal links is difficult when many parameters vary at once. We developed a simplified model of an elbow joint with two antagonistic Hill-type muscles, and varied the associated muscle parameters combinatorially over a large range. For a given parameter combination, we found optimal joint movements that minimized cumulative squared error to a target while starting and ending at rest. Emergent behaviour from the optimisations compared well to ballistic point-to-point arm movements in humans. Age-associated reductions of maximum isometric force, maximum strain rate and activation rate all had detrimental effects on performance, independent of other parameters. In contrast, deactivation time and passive parallel stiffness had no effect on performance on their own, but pronounced interactive effects with each other. Increasing stiffness reduced joint movement time at fast deactivation rates, but increased movement time at slow deactivation rates. This occurs because antagonist muscles resist the passive tension at rest, but are stretched eccentrically by the agonist, amplifying their active resistive force. Fast-deactivating muscles can avoid this resistive effect, allowing the passive stiffness to amplify accelerating force and enhance performance. In all cases, coactivation emerged as optimal during and after the braking period, and during the acceleration phase when stiffness increased. As deactivation time increased, so too did coactivation levels– but coactivation was not generally associated with a reduction in performance. Our simulations offer evidence that age-related changes in muscle strength, activation time and maximum contraction velocity can reduce ballistic performance in a goal-directed task, but the effects of increased muscle stiffness and deactivation time depend on their relative values.

**Data availability statement:** All code and data supporting this article are available at the following link: https://doi.org/10.5281/zenodo.15802371.

**Funding:** This work was supported by a Wellcome Trust Investigator Award 215618/Z/19/Z to CT Richards (https://wellcome.org). The funders played no role in the study design, data collection and analysis, decision to publish, or preparation of the manuscript.

**Competing interests:** The authors have declared that no competing interests exist.

## Author summary

As humans age, our muscles can get weaker, less excitable, stiffer and slower. At the same time, older adults tend to reach more slowly– making balance recovery more difficult. Since older muscles change in multiple ways simultaneously, we don't know which alterations most affect performance, or whether they interact. We developed a simple model of a human elbow joint, and used optimisation to find muscle activities that optimised speed and accuracy. We then digitally "aged" the muscle by modifying its properties to determine which most strongly alter performance. We found that the muscle activation rate, as well as its peak force and maximum contraction speed, most affect the speed and accuracy of reaching. On their own, muscle stiffness and speed of deactivation (turning "off" the muscle) don't affect performance. If the muscle turns off quickly, high stiffness helps the arm move faster, analogous to loading and firing a slingshot. However, if the muscle turns off slowly, higher stiffness slows the arm down– as if trying to shoot a slingshot without releasing the elastic bands. Altogether, our results help point to the key muscle changes that slow movement with ageing, allowing future research to better target therapeutic interventions.

## Introduction

Muscle is the dominant engine of animal movement, and physiologists have long been interested in how variations in muscle contractile properties link to functional outcomes [1–4]. Physiological changes in muscle are especially pronounced (and best studied) during ageing. As mammalian muscles age, they generally exhibit reductions in isometric tensile strength [5–10], slower activation and deactivation [5,10–14], reductions in the effective (muscle-level) maximum strain rate [8,9,15], and increases in passive parallel stiffness [6,10,16,17].

In addition to changes in muscle contractile properties, ageing is also associated with changes in locomotor behaviour and performance. Elderly humans tend to reach more slowly and with more effort [18,19], with less precision [18], and with higher levels of cocontraction [20,21]. Performance reductions in reaching tasks are associated with negative health outcomes, particularly fall risk. Older adults rely more on arm reactions for balance recovery, but are slower to initiate and execute reach-to-grasp [22]. Establishing links between physiological changes in muscle and performance outcomes provides a theoretical basis for future research towards targeted treatment to maintain locomotor performance throughout life. However, direct links are difficult to establish, as ageing can involve multifactorial musculoskeletal and neurological changes [23] that are difficult to fully control for in empirical studies. Which changes to muscular parameters actually lead to performance differences in everyday movement? Which are most important, and do they interact?

To address the above questions, dynamic biomechanical models can be used to explore the effects of varying parameters in a well-defined control task.

Applying optimal control to these models yields behavioral predictions of motor control, providing an upper bound on performance for a given set of parameters. [4,24,25]. Despite their potential predictive power [24,26,27], highly complex models can be difficult to analyse in detail [27,28], as the number of interdependent parameters becomes intractably large. In contrast, simplified models can search a large number of parameter combinations to unravel links between underlying parameters, their interactions, and their consequences for optimal behaviour [29–33]. They also allow us to better understand the mechanisms between muscle change and performance outcomes [34]. As single-joint and multi-joint arm movements appear to use similar control strategies [35–37], we focus on a simplified point-to-point reaching movement about the elbow, allowing us to better understand the mechanisms underlying performance changes.

To that end, we propose a simplified two-muscle Hill-type model of a single joint. We assign parameters based on the human forearm and use optimal control to minimise time-integrated squared position error in a point-to-point movement. This metric of performance encapsulates both the time to reach the target, and rapid stabilisation around it. Using a large parameter sweep of isometric force, maximum strain rate, parallel passive stiffness, and activation and deactivation rates, we determine best-case performance outcomes for thousands of parameter combinations. The results of this investigation tell us which age-related changes might be associated with performance deficits, how and when they interact, and uncover causal links between muscle physiology and performance outcomes.

## Materials and methods

### Model design

We simulated a motor control task to extend a rotational joint (the elbow) from one point to another, starting and ending at rest, while minimizing a performance cost within a set time. The cost is given as the time-integrated squared error from the target (endpoint) position; more time spent away from the target accrues higher cost, and so high performance (low cost) indicates a solution that can rapidly reach and come to rest at the target. Although we choose model parameters to approximate a human elbow joint, the model could be easily extended to other single-degree-of-freedom joints through judicious selection of parameters.

The arm, muscle and activation models are based on [34,38], but considering only single degree-of-freedom flexion and extension about a frictionless elbow joint. This further simplification enables a far more thorough sweep of parameters than typically performed in musculoskeletal modelling studies. In order to keep the model simple and conceptual, the flexor and extensor muscles are identical in size, strength, muscle properties, and moment arms. The load they move is a lumped inertia equivalent to a combination of modelled forearm and hand inertias, with a fixed wrist. The motion is in a horizontal plane, such that there are no external forces. At 90° flexion, both flexor and extensor are at optimal (resting) length, and the forearm is perpendicular to the upper arm (Fig 1).

We first define the point-to-point movement as an optimal control problem (OCP). Using the above elbow model, we discretise and solve this OCP for a given set of parameters. The parameters of interest are activation and deactivation time constants ($\alpha_a$, $\alpha_d$), peak isometric force ($F_{max}$), maximum contraction velocity ($v_{max}$), and passive stiffness ($c_1$). Starting from a baseline parameter set based on [34], we compare kinematic and excitation predictions to empirical data of healthy elbow motion as a validation. Next, we choose parameter ranges based on ageing literature to explore the effects of variation. We then perform full factorial parameter sweeps, solving the optimal control problem for every combination of parameter values over all their ranges simultaneously. This generates a dataset of performance metrics as a response with parameter values as independent predictors. To explore the relationships between these predictors, their interactions, and performance, we regress the data set onto a linear model including first-order interaction terms, and determine which parameters (or their interactions) have the strongest effect on performance.

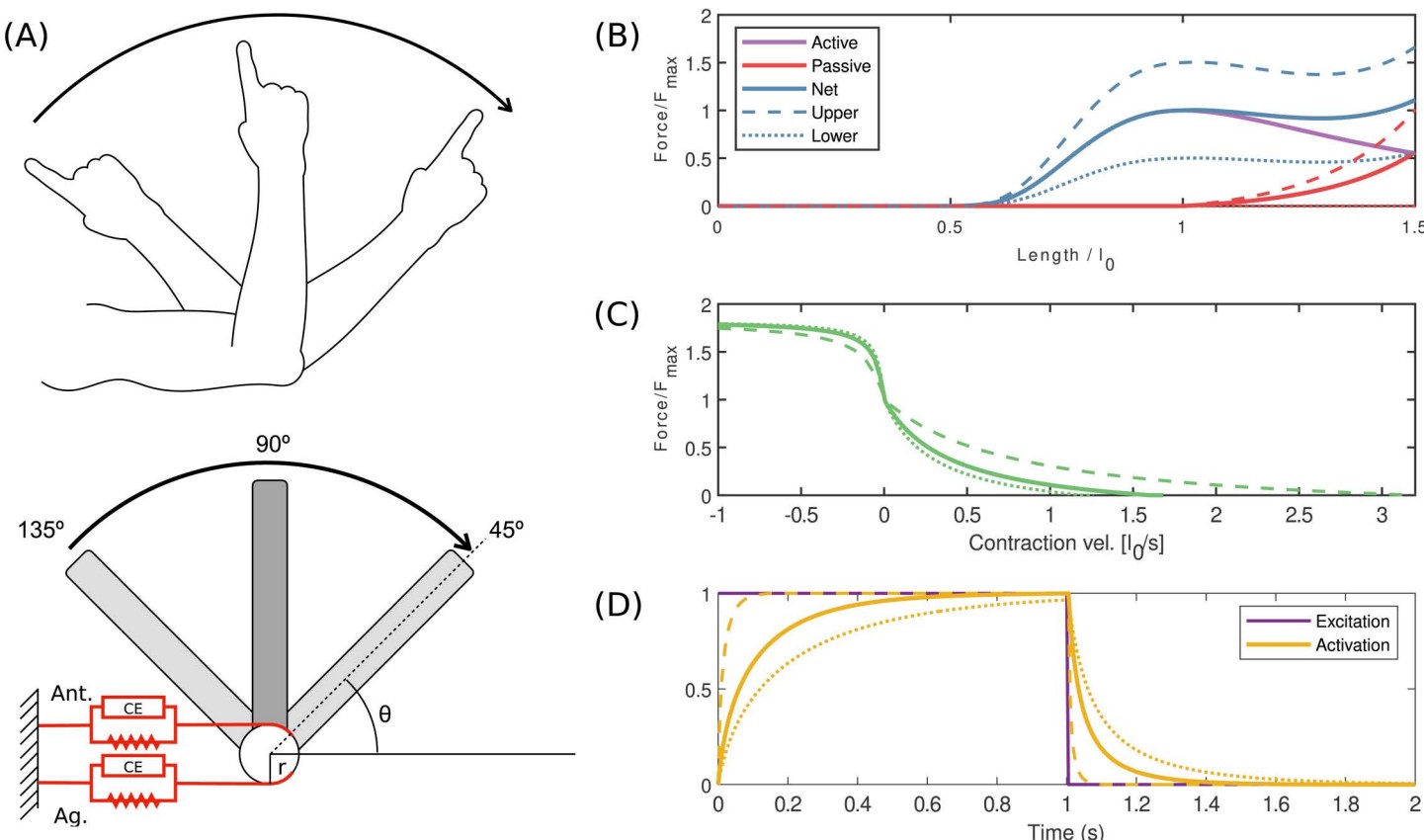

**Fig 1. Schematic showing the modelling approach. (A)** Single joint model. Muscles are Hill-type with constant moment arm **r**. **(B-D)** Hill-type muscle and activation properties. Baseline case is indicated as a solid line, while the lower and upper ranges explored in the parameter sweep are shown as dotted lines. Parameters are specified in Tables 1 and 2. **(B)** Hill-type force length properties. Active force length is based on [39] and parallel passive force length based on [34]. **(C)** Hill-type force-velocity properties based on [39]. **(D)** Activation model based on [40]. Mathematical representations are smoothed to remove discontinuities for trajectory optimisation; for details, see S1 Appendix.

## Trajectory optimisation

For the current study, the optimisation problem is to start at a given position at rest, and choose as controls time-varying excitation values that minimize the sum of squared position error (SSE) to a target, while satisfying dynamic constraints, and finally ending at rest. SSE was defined as

$$J_{SSE} = \sum_{i=1}^{N} (l_{ag,i} - l_{ag,target})^2 + (l_{ant,i} - l_{ant,target})^2$$

(1)

where $l_{ag,i}$ and $l_{ant,i}$ are muscle lengths at timestep $i$, and $l_{ag,target}$ and $l_{ant,target}$ are muscle lengths at the target position, for the agonist and antagonist, respectively. For comparison to empirical EMG data, start and target elbow flexion positions of 120° and 60° respectively were used. For parameter sweeps, start and target positions of 135° and 45° were used (Fig 1A). As the end position was not constrained, the objective was augmented with an additional term penalizing an end position far from the target

$$J_{end} = w \cdot \left( (I_{ag,N} - I_{ag,target})^2 + (I_{ant,N} - I_{ant,target})^2 \right) \tag{2}$$

where the weighting $w = 0.01$ and $N$ corresponds to the final time step. The final objective was a combination of terms,

$$J = J_{SSE} + J_{end}. \tag{3}$$

The time horizon $T$ was set as 0.4 s, chosen such that the worst-performing case could reach the target.

Force-length and force-velocity relationships are Hill-type following [39]. These were smoothed where cusps and discontinuities existed (see S1 Appendix), so that exact derivatives could be calculated using CasADi [41]. Excitation-activation coupling was set using a first-order model following [40]. Muscle excitations were set as piecewise constant controls. Modeling constants and baseline parameters (Tables 1 and 2) are chosen to match [38], with the exception of $b_1$ in the force-length relationship. This was given the value of 2 so that the Hessian of force to length was finite everywhere. The maximum eccentric normalised force (parameter $d_2$) was scaled to changes in $F_{max}$ in order to maintain absolute eccentric force ($d_2 = 1300 \, \text{N} \, F_{max}^{-1}$), in accordance with evidence that eccentric strength remains relatively well preserved with advancing age in humans [42].

The optimisation problem was discretised into 5 ms timesteps, and transcribed with the CasADi opti stack using multiple shooting with 4th-order Runge-Kutta trapezoidal integration. Excitation inputs were piecewise constant over these

**Table 1. Modelling constants.**

| Parameter | Symbol | Value |
|---|---|---|
| Muscle moment arm | $r$ | 0.04 m |
| Muscle resting length | $l_0$ | 0.32 m |
| Simulation duration | $T$ | 0.4 s |
| Elbow resting angle | $\theta_0$ | 90° |
| Initial and final elbow flexion | $[\theta_i, \theta_f]$ | [135°, 45°]† |
| Arm moment of inertia about elbow | $I$ | $6.88 \times 10^{-2}$ kg m² |
| Force-length shape parameters | **b** | [2, -1.3, 0.53] |
| Passive parallel stiffness parameters | **c** | $[c_1, 5, c_3]$* |
| Slack length per $l_0$ | $c_3$ | 1.0 |
| Force-velocity shape parameters | **d** | [4, 1.4, 30.24] |
| Smoothing parameter | $s$ | 500 |

*For variable values of $c_1$, see Table 2 † Excursion for the parameter sweeps. For comparison to empirical data, a range of [120°, 60°] was used.

**Table 2. Baseline parameters and values tested during parameter sweeps. For the finer-grid parameter sweep, deactivation time and stiffness values were varied over the same range below, in 0.6 ms and 0.005 unit increments respectively, while all other values were held at baseline.**

| Parameter | Symbol | Baseline value | Range tested (× baseline value) |
|---|---|---|---|
| Peak active force | $F_{max}$ | 1300 N | [0.5, 0.8, 1.0, 1.5] |
| Maximum strain rate | $v_{max}$ | 1.6 $l_0$ s⁻¹ | [0.75, 1.0, 1.5, 2.0] |
| Activation time constant | $\alpha_a$ | 94.5 ms | [0.15, 0.25, 0.5, 0.75, 1.0, 1.5, 2.0] |
| Deactivation time constant | $\alpha_d$ | 65 ms | [0.15, 0.25, 0.5, 0.75, 1.0, 1.5, 2.0] |
| Stiffness constant | $c_1$ | 0.05 | [0, 0.6, 1, 1.4, 1.8] |

timesteps, constrained between [0, 1] (but otherwise free to be selected by the optimisation routine). The trajectory optimisation problem was solved using IPOPT [43] with the MA57 linear solver [44]. For every case in the large parameter sweep, as well as for the baseline case, the initial guess consisted of a linear interpolation of muscle lengths from the starting value at $t = 0$ to the target value at $t = T$, and zero values for all other states and controls. For the finer-grid parameter sweep, the baseline case solution was used as the initial guess (since other parameters were held at baseline values).

The optimisation proceeded for every parameter combination outlined in Table 2, a total of 3920 trials, as well as for the 399 trials of the finer-grid sweep, all of which converged to unique optimal solutions. The performance of all solutions was evaluated as the Root Mean Square Error, defined as

$$J_{RMSE} = \frac{1}{2l_0} \sqrt{\frac{J_{SSE}}{N}},$$

(4)

where division by 2 allows $J_{RMSE}$ to represent the normalised error of one muscle (as both muscles are symmetric and contribute equally to $J_{SSE}$).

## Parameter range selection

Baseline parameter values were taken from [34]. Parameter ranges to explore were based on putative changes with age observed in empirical studies. For peak tetanic force, Brown *et al.* [6] found at most a 2-fold reduction for healthy older mice compared to young controls (in the plantaris muscle), and a 5-fold reduction for "sendentary" (hindlimb offloaded) mice. We therefore considered a 3-fold range in peak tension (0.5 to 1.5 times the baseline value). The same study found at most a 2.5-fold increase in normalised passive parallel stiffness for old mice compared to young, and at most a 5.5 fold increase in hindlimb-offloaded old mice compared to young. We chose a maximum stiffness 1.8 times the baseline value (higher values led to difficulties in simulation convergence) and also considered the case of no muscle parallel stiffness.

Raj *et al.* [15] compiled empirically measured changes in muscle $v_{max}$ values in humans, finding they are reduced by at most 1.6 times in elderly subjects compared to young. However, an empirical study of human elbow flexion [8] reports $v_{max}$ values twice as large as used by Murtola and Richards [34]. To account for this uncertainty in $v_{max}$ variation, we chose a range from 0.75 to 2 times the baseline value (a 2.7-fold change).

Doherty *et al.* [45] reviewed studies on muscle contractile properties with age, finding that the largest reported change in time to peak tension in older populations was at most 1.5 times the younger value in the extensor digitorum brevis (EDB), and about 1.25 times the younger value in triceps surae (TS). Time to half relaxation increased 2 times in the EDB and up to 1.2 times in the TS. However, it is challenging to map twitch-based activation parameters to a first-order model that does not reproduce twitches [38], leading to uncertainty as to the most representative numerical value. Moreover, we noted parameter interactions with deactivation that warranted further investigation. Therefore, we chose a much larger variation in activation and deactivation time (from 0.15 to 2 times the baseline value) to explore these issues, while acknowledging this is much larger net change than could be reasonably expected during human ageing.

All parameter ranges were discretized (Table 2), and every combination of parameters was input into the optimisation model to find trajectories that minimised the objective (Eq 3). In addition to the large parameter sweep, which varied all parameters over their full ranges outlined above as well as every combination, we also performed a finer-grid parameter sweep, which varied only deactivation time and stiffness constants. These were varied over the same range as in the larger parameter sweep, but in 0.6 ms and 0.005 unit increments, respectively. For the finer parameter sweep, all other parameter values were held at baseline.

## Regression

In order to determine which of the parameters, or their interactions, contribute to changes in performance in the dataset generated by the large parameter sweep simulations, we performed linear regression of $J_{RMSE}$ against the five muscle

parameters and their interactions. A common method for determining the relative importance of individual terms in a regression model is "standardized betas" [46], where predictors are standardized by subtracting their means and dividing by standard deviation. The regression coefficients for each term thus represent the change in the response variable from a unit (standard deviation) change in the predictor, and thus can be seen as the relative strength of the predictor in affecting the response [46].

However, the standardized beta method assumes a Gaussian distribution of the data, whereas the predictors in this case are not randomly distributed. Instead, we normalise each muscle parameter of interest and the response variable ($J_{RMSE}$) by subtracting its minimum value and dividing by its range. Thus, the regression coefficient for each term represents the fractional change in $J_{RMSE}$ expected by changing the predictor from its minimum to maximum value (in the absence of interactions).

A criticism of standardized betas is that results can be misleading if predictors are strongly correlated with each other [46]. However, the predictors in this deterministic model are uncorrelated by design, and so regression coefficients should be a strong indicator of importance. Nevertheless, if different predictor ranges were used (*e.g.,* doubling the range of $F_{max}$ in the parameter sweep), then different regression coefficients could emerge, leading to different measures of importance. This issue is beyond the scope of the current study, but a critical consideration when interpreting results.

Coactivation at a given time $t$ was defined as the minimum activation between the antagonist and agonist at a given timepoint $\left(a_c(t) = \min(a_{ant}(t), a_{ag}(t))\right)$, while the average of coactivation was taken over the duration of the simulation,

$$\bar{a}_c = 1/T \int_0^T a_c(t)\, \mathrm{d}t$$

(5)

## Results

Using baseline parameters (Table 1), predictive simulation of optimal elbow extension exhibits qualitative similarities to empirical ballistic data from [47] (Fig 2). In both, the elbow extension profile is smooth and settles at the target after overshooting. The simulation predicts two main features that also appear in experimental data: (a) a triphasic pattern of excitation, where agonist and antagonist excitation are out of phase in an agonist-antagonist-agonist pattern [48,49], although the simulated excitation and empirical EMG are offset in time (Fig 2); and (b) an additional final burst of antagonist excitation following the completion of the movement, which corresponds to a sustained level of excitation in the agonist and antagonist. Rapid single-joint arm movements are associated with relatively high levels of tonic coactivation following movement [50].

When input parameters change from baseline, differences in optimal trajectories emerge, although the general pattern remains the same. In the baseline case, muscle length overshoots the target before rapidly settling to the target value (Fig 3A, red line). With the "youngest" parameter combination (highest $F_{max}$ and $v_{max}$, and lowest $\alpha_a$, $\alpha_d$ and $c_1$) as well as the "oldest" (lowest $F_{max}$ and $v_{max}$, and highest $\alpha_a$, $\alpha_d$ and $c_1$) the same pattern emerges, but with about 2/3 and twice the time to reach the target, respectively, and no overshooting in the "oldest" condition. The velocity profiles resemble a skewed "bell-shape" [52], with peak velocity occurring late in the propulsive phase (Fig 3B). Velocity fluctuations during stabilisation increase from the "older" case, to baseline, to the "younger" case. Agonist activation increases early in the movement, and begins deactivating as antagonist activation begins (Fig 3C), reflecting the triphasic pattern. The "youngest" condition achieves full activation, and exhibits higher compensatory antagonist activation. In the baseline case, and even more so in the "oldest" case, peak activation is smaller. Agonist activation leads to an initial rapid increase in agonist force (Fig 3D), which decays due to the increase in contractile velocity. When muscle length is close to the target, the antagonist produces a large braking force, well above the agonist peak force, which is then followed by an increase in agonist force as

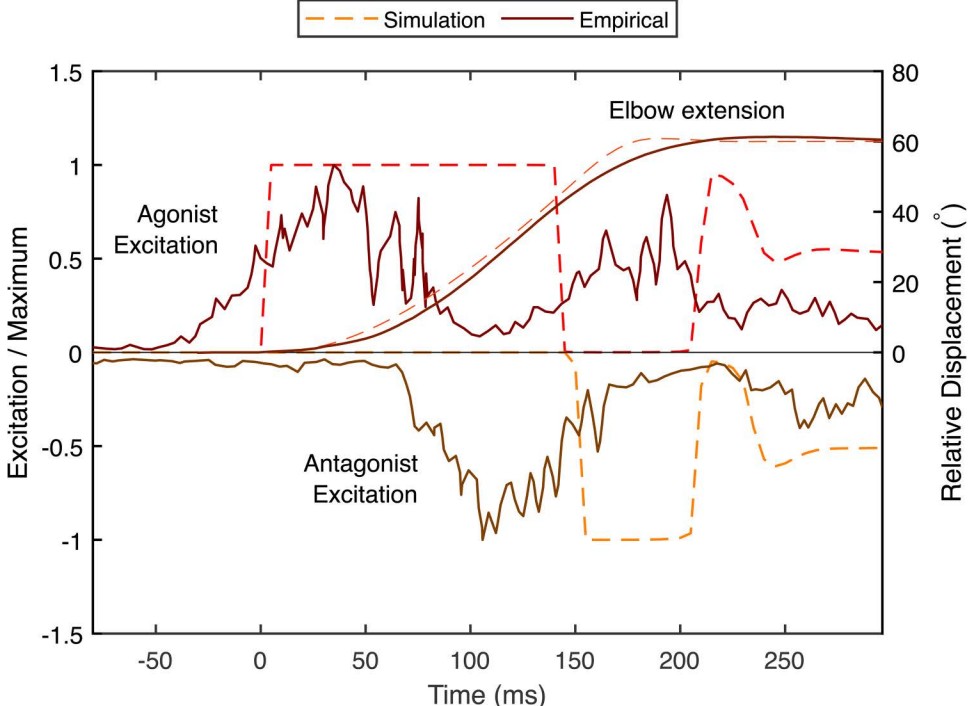

**Fig 2. A predictive simulation with baseline parameters exhibits qualitative similarities to empirical EMG data from [47].** Both exhibit a smooth arm angle profile with similar maximum velocity, and a triphasic pattern of muscle excitation: a peak in the agonist excitation is followed by a peak in the antagonist and a subsequent peak in the agonist. In addition, both model and empirical data have a sustained level of co-excitation after the arm has come to a stop. EMG peaks occur earlier than those predicted by the model, and have a relatively smooth rise and fall. Empirical data were extracted using WebPlotDigitizer [51]. Note that the empirical and simulated data represent a 60° elbow excursion, as opposed to 90° in the remaining simulated data.

the system slows down. Both agonist and antagonist settle into an equal and opposite sustained force until the end of the simulation time (0.4 s).

A performance landscape across the range of parameter combinations is shown in Fig 4. In general, parameters move from "younger" to "older" values from the top left to bottom right– in other words, some combination of increase in parallel stiffness (upper four panels to lower four panels), decrease in $v_{max}$ (top row to bottom row for a given stiffness), decrease in force (left to right column of panels), or increase in deactivation or activation time (downward or rightward in a single panel, respectively). As parameters shift to "older" values, performance generally decreases– corresponding to an increase in root-mean-square-error (RMSE). However, there are two notable exceptions. In the absence of stiffness (Fig 4A; panels i-iv), changing deactivation time appears to have no effect on performance (indicated by vertical contour lines). When stiffness is appreciable (Fig 4B; panels v-viii), increasing deactivation time decreases performance. However, for any point in A, the corresponding point in B may have relatively higher or lower RMSE, depending on its location.

This phenomenon is illustrated in Fig 5, showing a performance landscape as a function of deactivation time and stiffness; all other parameters are held constant at baseline values. A hypothetical boundary across the parameter space emerges (Fig 5, white solid line). When deactivation time values are near this boundary, stiffness has little to no effect on performance. When deactivation time is higher than these values (below the line), increasing stiffness decreases performance (indicated by an increase in RMSE). When deactivation time is lower (above the line), increasing stiffness *increases* performance. The total change in RMSE from deactivation and stiffness alone (Fig 5) represents 10.6% of the

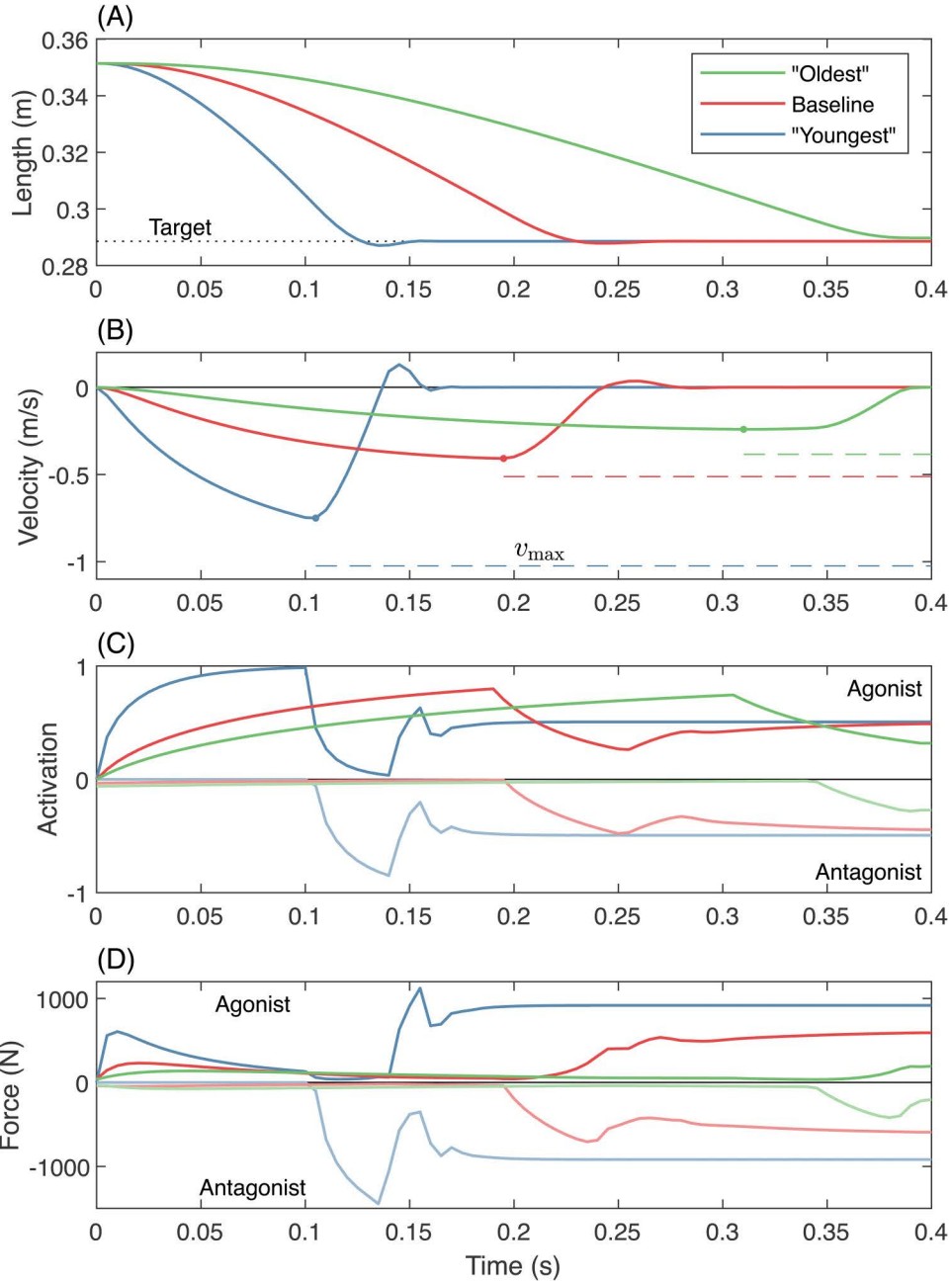

**Fig 3. Time series results of optimal solutions with varying input parameters.** The "oldest" condition (green) represents muscle with maximal stiffness ($c_1 = 0.09$), maximum activation and deactivation time constants ($\alpha_a = 189$ ms, $\alpha_d = 130$ ms), minimal peak force ($F_{max} = 650$ N), and minimal peak strain ($v_{max} = 1.2\ l_0\ s^{-1}$); *i.e.,* the bottom right corner of the bottom right panel in Fig 4. The "youngest" condition (blue) represents the opposite extreme; the top left corner of the top left panel in Fig 4, with $c_1 = 0$, $\alpha_a = 14$ ms, $\alpha_d = 10$ ms, $F_{max} = 1950$ N and $v_{max} = 3.2\ l_0^{-1}$. The baseline case (red) represents intermediate values, outlined in Table 2. Note that "youngest" and "oldest" represent extremes explored in the parameter sweep, and not necessarily the expected change in muscle properties through a person's lifetime. **(A)** Agonist muscle length reaches the target in all cases, but more quickly for the "younger" muscles. **(B)** Agonist contraction velocity increases more rapidly in the "younger" muscle. The peak contraction velocity (dot marker) is closest to $v_{max}$ (dashed line) in the baseline case. **(C)** The "youngest" agonist rapidly approaches peak activation, whereas the "older" muscles reach lower levels of peak activation before deactivating for the braking phase. Antagonist muscles ("negative" activation) show similar patterns. **(D)** "Younger" muscle displays higher levels of acceleratory and braking force, as well as higher levels of sustained cocontraction after stabilisation.

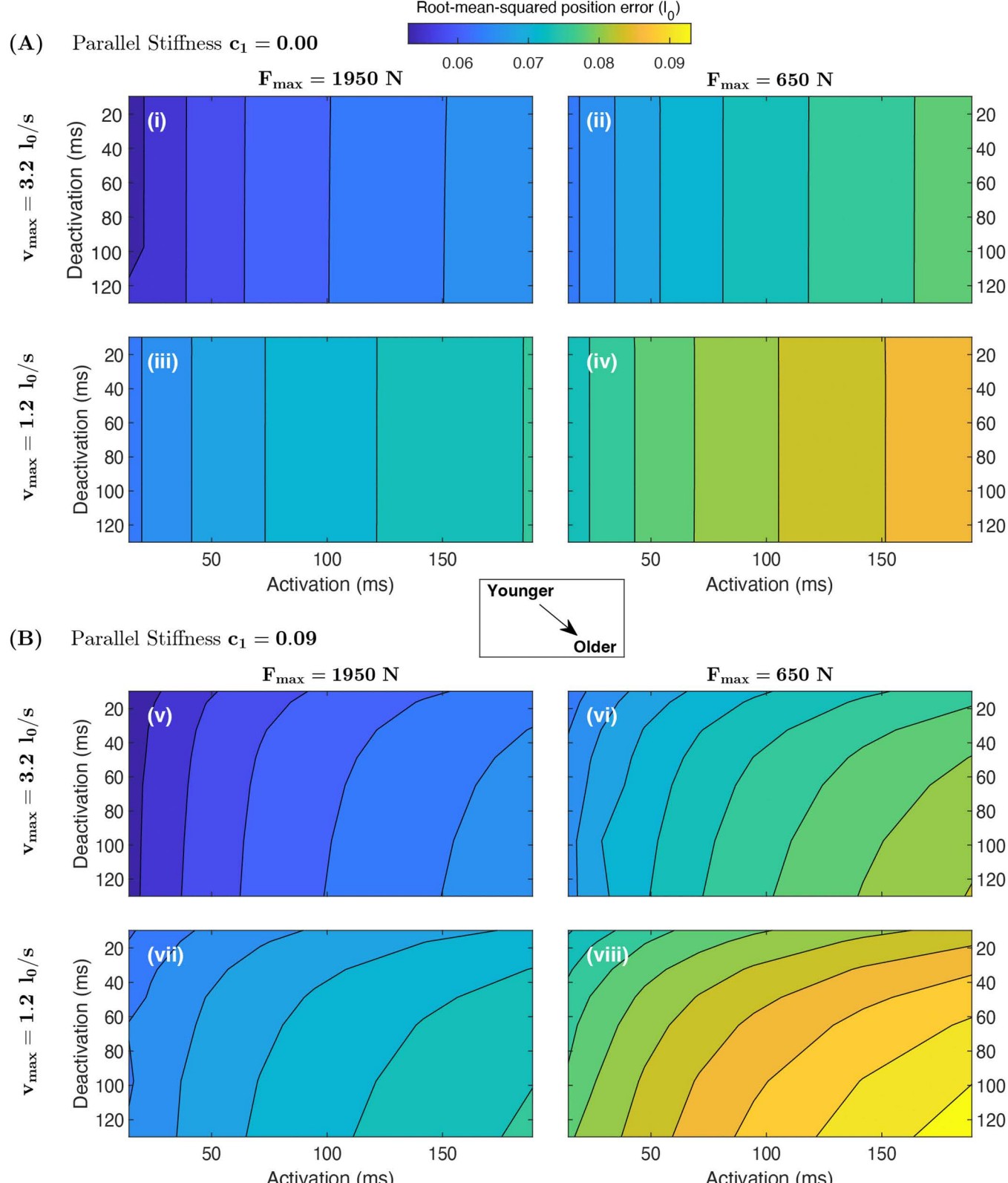

**Fig 4. Contours of total Root-Mean-Squared Error (RMSE) during ballistic arm pointing tasks show generally decreased performance as parameters shift towards "older" values, and an interaction between deactivation rate and stiffness.** Each panel shows RMSE as a function of

deactivation and activation time (increasing to the bottom right), for some combination of maximum strain rate ($v_{max}$), peak active tension ($F_{max}$) and passive parallel stiffness ($c_1$). The top four panels (i-iv) are for no passive parallel stiffness, and the bottom four (v-viii) for high stiffness ($c_1 = 0.09$). Within each stiffness condition, $F_{max}$ decreases from the left column to right column of panels, and $v_{max}$ from the upper row to the lower row. Altogether, the plots are arranged such that moving from top left to bottom right indicates a shift from relatively "young" to relatively "old" muscle. **(A)** If the muscle has no passive stiffness ($c_1 = 0$), RMSE increases (*i.e.*, performance is reduced) when activation time increases, $F_{max}$ decreases, and/or $v_{max}$ decreases, but the vertical contour lines indicate that performance does not vary with changes to deactivation time. **(B)** When the muscle has high stiffness ($c_1 = 0.09$), RMSE also increases with deactivation time. However, relative to the same parameter combinations without muscle stiffness (the equivalent point in A), the RMSE can either decrease (rightward shift in contour lines at low deactivation) or increase (leftward shift in contour lines). RMSE varies from 0.0534 $l_0$ (top left corner of i) to 0.0943 $l_0$ (bottom right corner of viii), corresponding to a three-fold increase in time to target (from 0.12 to 0.38 s, S2 Fig). Parameter ranges are larger than the expected natural variation with age, especially in activation and deactivation; see Methods for details.

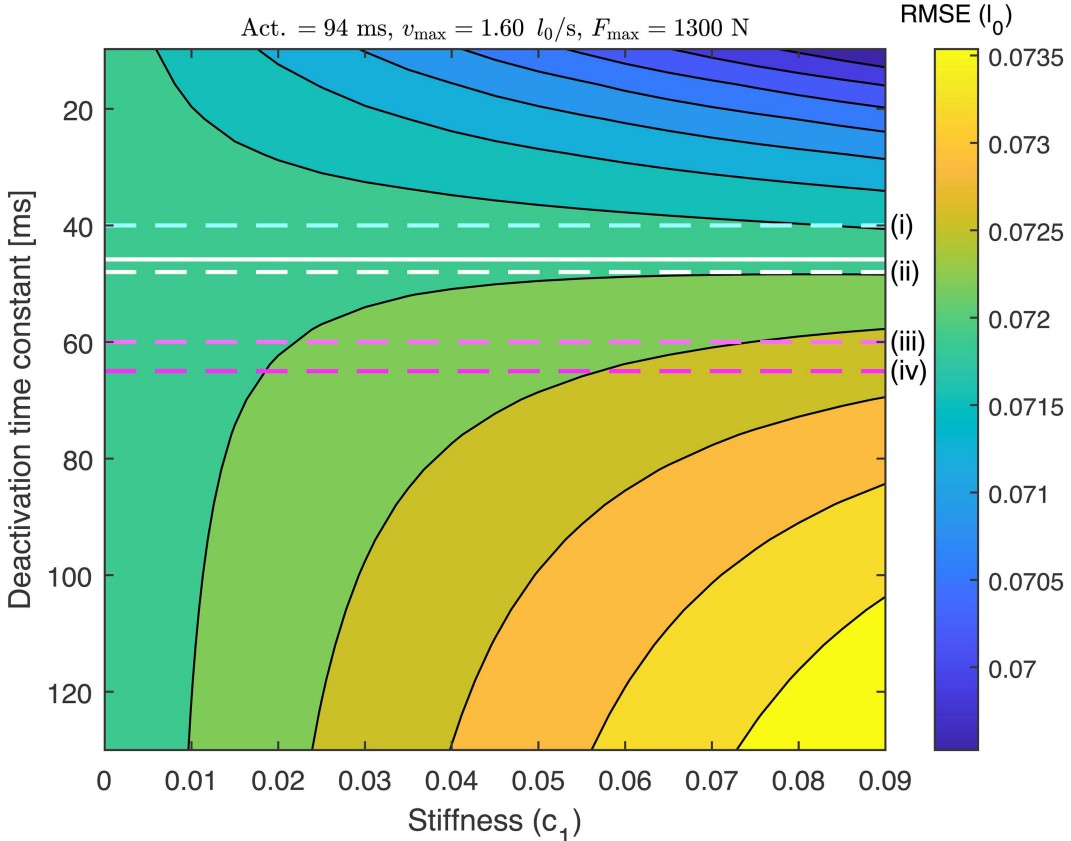

**Fig 5. Root Mean Squared Error (RMSE) contours as a function of deactivation time and passive parallel stiffness, with other parameters held at baseline values.** When the deactivation is fast (upper portion of plot; small time constant), increasing stiffness decreases RMSE and enhances performance. When the deactivation is slow (large time constant), increasing stiffness decreases performance. Intermediate values of deactivation exist where stiffness changes have little to no effect; at deactivation time of 46 ms (white line), a 0.01 unit change in $c_1$ changes RMSE by less than $10^{-4}$ $l_0$. Various studies use this same activation model in dynamic simulations of ageing, but with different deactivation time constants (dashed lines)– potentially leading to different effects of passive stiffness. **(i)** The default OpenSim value (*e.g.,* used by [53]) is 40 ms, implying an increase in performance with increased stiffness. **(ii)** Nowakowski *et al.* [54] used an aged value of 48 ms in their simulation, which would lead to no stiffness effect in the current study. **(iii)** Thelen [40] used 60 ms and **(iv)** Murtola and Richards [38] used 65 ms in their aged models. These values would lead to a decrease in point-to-point performance with increased stiffness, as found by Murtola and Richards [34] with a related model. The total performance change represented in this plot (0.0043 $l_0$) represents 10.5% of the maximum variation seen over the entire performance landscape (Fig 4), and corresponds to a 17% increase in time to target (from 0.205 to 0.240 s; see also S2 Fig).

maximum performance change seen over the entire parameter sweep (Fig 4), and corresponds to an increase in time to target from 0.205 to 0.240 s (a 17% increase).

In the presence of stiffness, muscle deactivation affects the net accelerative torque about the joint. In the initial posture, the agonist is stretched, increasing its passive contribution (Fig 6A, $t = 0$). To maintain stasis prior to joint movement onset, the antagonist must supply an equivalent active force (Fig 6B). As the agonist contracts and the elbow extends, the antagonist becomes stretched, experiencing force amplification as it shifts to the eccentric part of the force velocity curve. If the antagonist can "switch off" quickly, it can reduce its resistive force below the agonist's passive contribution. In this case, the net extending torque about the joint is larger than what it would be without any passive stiffness (Fig 6C, cool colours), and the passive stiffness has a net benefit to performance. However, if the antagonist cannot switch off quickly, then its force rises *above* the agonist's passive contribution (Fig 6B, warm colors), and the net extensor torque is reduced compared to the no-stiffness case (Fig 6C, warm colours).Because the antagonist is initially active, coactivation is high at

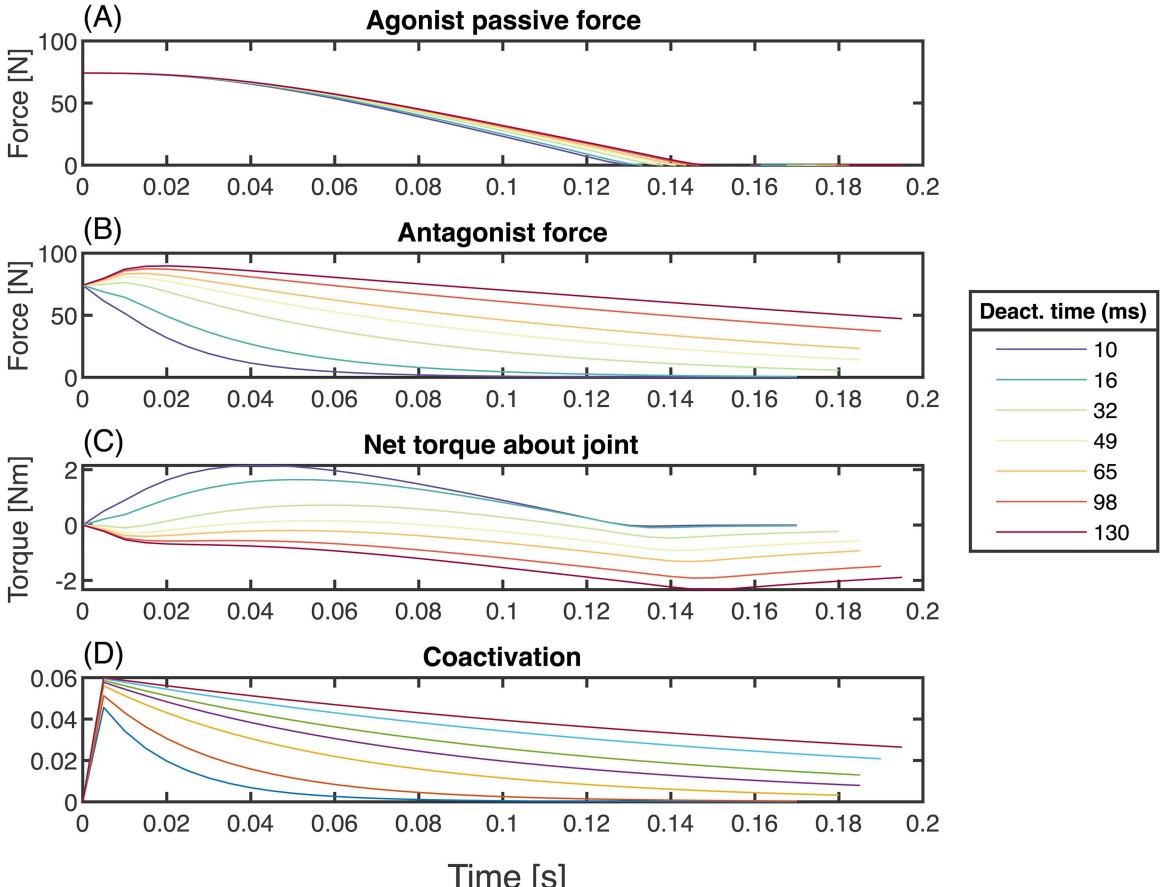

**Fig 6. Deactivation time affects the net force about the joint in the presence of agonist passive parallel force.** Shown here are results with baseline parameters in all but deactivation time, and $c_1 = 0.09$. **(A)** Agonist passive force is positive at $t = 0$ as the muscle is stretched, generating a positive torque about the joint. **(B)** The antagonist must produce an equal and opposite active force to keep the joint at rest at $t = 0$. As the arm begins to accelerate, the antagonist is stretched, moving it to the eccentric part of the force velocity curve (Fig 1C). If the muscle can deactivate quickly (cooler lines), its force can rapidly drop. However, if the muscle deactivation is slow (warmer lines), the antagonist's force rapidly increases in excess of the extra passive force from the agonist. **(C)** The net added torque at the joint, (*i.e.,* the sum of antagonist and passive agonist force) thus depends on antagonist deactivation time. If deactivation time is low, the system gets an accelerative boost from the agonist's passive force. If the deactivation time is high, the antagonist's resistive torque exceeds the passive torque, and the additional net torque is negative. **(D)** Early-onset coactivation occurs regardless of deactivation time. However, coactivation persists at higher levels when deactivation time is higher (warmer lines).

the beginning of the acceleration phase and decreases in time, more quickly for fast deactivation (Fig 6D). Coactivation also increases with deactivation in the absence of stiffness, but only in the the braking phase. Fig 7 shows the results of changing deactivation while keeping stiffness zero and other parameters at baseline. The agonist muscle strain trajectories are virtually indistinguishable (Fig 7A), with only slight differences in oscillation around the target point while the system comes to a stop. The strain achieves similar peak velocities (Fig 7B), with lower deactivation times resulting in higher-amplitude, higher-frequency oscillations about zero during the braking phase. The onset of agonist deactivation occurs slightly earlier when deactivation time is larger (15 ms earlier for the highest *versus* lowest $\alpha_d$), and the antagonist begins activating slightly earlier as well (at most 5 ms). As the arm slows down, antagonist deactivation remains high when deactivation is slower, and the agonist increases its activation for longer. This results in larger sustained forces in both muscles (Fig 7D), leading to higher levels of cocontraction both during the braking phase and once the simulation has terminated (Fig 7E). All deactivation conditions have sustained levels of equal activation and force in both muscles at the end of the motion.

In order to determine the biggest contributors to performance in this dataset, we performed linear regression of RMSE against the five variables and their interactions plus a constant (a total of 16 terms). Best-fit coefficients are shown in Table 3, where the magnitude of the coefficient represents how much changing the specific term from its minimum to maximum value (holding all other terms and interactions constant) changes RMSE on average (as a fraction of total possible change). By analogy to the "standardised beta" statistical technique [46], we define these coefficients as the "importance" of the term. Five of the 16 coefficients are important to first order (magnitude > $10^{-1}$). One is the intercept value, and three others are the individual parameters of activation rate, maximum strain rate and maximum isometric force; deactivation rate and stiffness are not important on their own to first order. The only important interaction effect (to first order) is that of deactivation and stiffness together.

To further check whether these first-order terms account for the vast majority of the variation, we fit a linear model with only these terms (Table 3). The fit has $R^2 = 0.94$, 1% smaller than if all 16 terms are included, indicating that first-order terms are sufficient to explain most of the variation in performance. A fit further removing the interaction term yields $R^2 = 0.92$.

We also performed linear regression of $J_{RMSE}$ against mean coactivation in each simulation, to see whether coactivation was associated in any way with performance. Mean coactivation ($\bar{a}_c$, Eq 5) ranges from 0.034 to 0.415. We find that mean coactivation is strongly correlated with performance ($R^2 = 0.7$), and that higher coactivation is associated with better performance ($J_{RMSE}/\max(J_{RMSE}) = -0.89\,\bar{a}_c + 0.93$, $p < 10^{-6}$).

## Discussion

The aim of this study is to determine how age-related changes to muscle contractile properties affect ballistic reaching performance in a point-to-point task. Here performance is defined as the cumulative root-mean-square distance from the arm to the target (Eq 4), and is driven by how quickly the arm can both reach and converge to its target point. When control is optimised to maximise performance, it resembles rapid point-to-point movement in human elbow flexion (Fig 2; [47]), including displacement trajectories that are similar to empirical data in shape and duration, and a triphasic excitation pattern– a well-documented phenomenon in ballistic movements [48,49]. While the simulation is not an exact simulacrum of human behaviour– for example, the excitation peaks come later than empirical EMG signals– the general match to empirical patterns (with no fitting to data) shows that the simplified model captures salient features of rapid targeted joint excursions. By systematically changing parameters and re-solving the optimisation problem, we constructed a performance landscape as a function of muscle parameters to determine how shifts in muscle characteristics affect optimal performance (Fig 4).

We find that reductions in peak isometric force and maximum strain rate– all associated with ageing muscle– each lead to decreases in performance, and that these effects do not interact. Increases in both deactivation time and stiffness– also

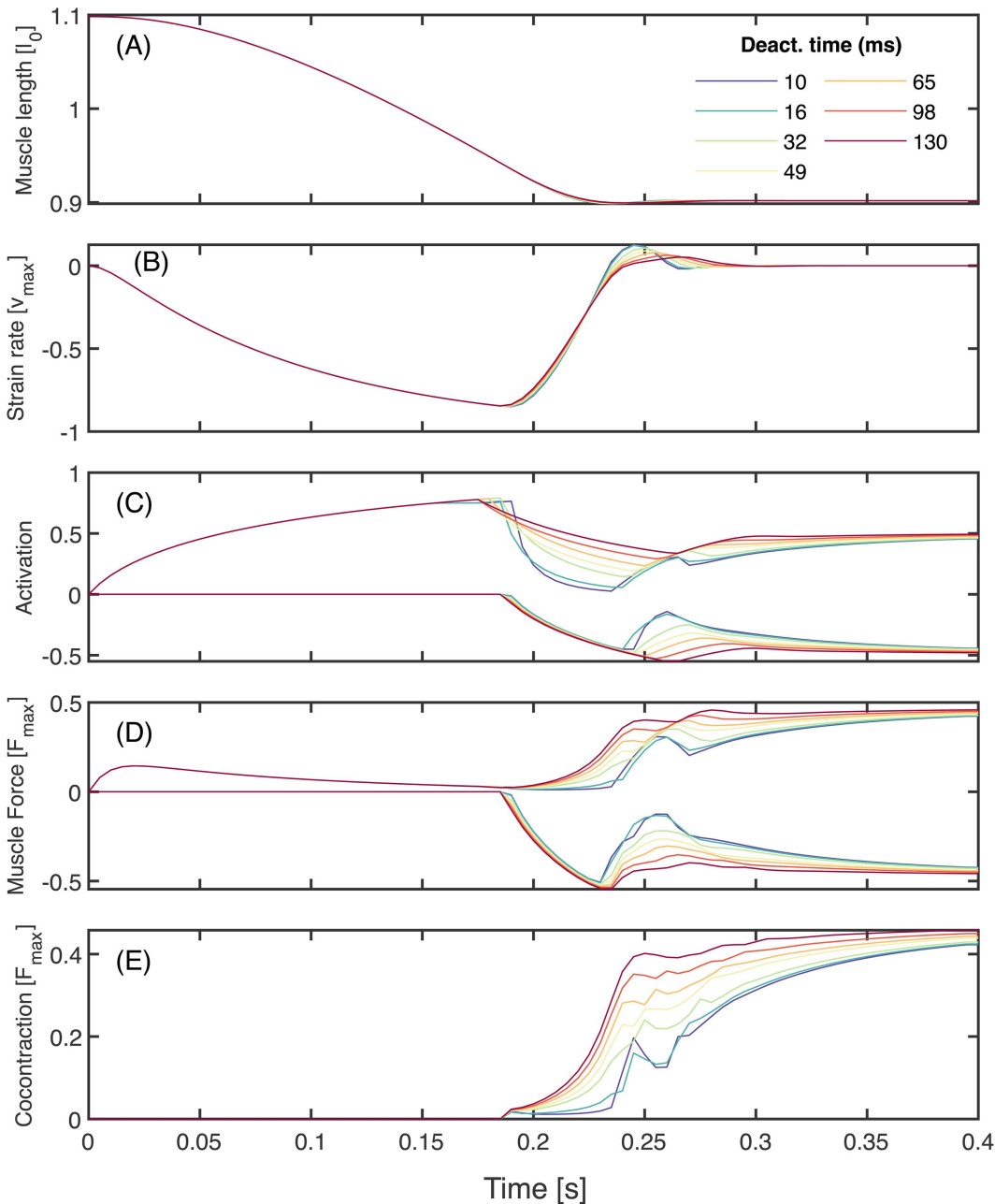

**Fig 7. Slowing deactivation increases the level of cocontraction in optimal ballistic movements at zero parallel stiffness, without affecting performance.** Here all other parameters are held constant at baseline values (Table 1). **(A)** Agonist muscle length trajectories appear nearly identical for all deactivation times. **(B)** Slight differences in agonist muscle velocity emerge during the deceleration and stabilisation portion of the trajectory. Muscle contraction velocity approaches $v_{max}$ in all cases. The antagonist strain and strain rate are equal and opposite to the agonist's at every instant. **(C)** Positive activation corresponds to the agonist, while negative corresponds to the antagonist. When deactivation time is low (cool colours), the agonist rapidly decreases activation as the antagonist activation rises. When deactivation time is high (warm colours), the agonist cannot deactivate fast enough, and the antagonist exhibits a corresponding higher level of peak activation, resulting in higher sustained levels of activation post stabilisation. **(D)** Because the agonist's muscle velocity approaches $v_{max}$, the agonist's force is small immediately before the target is reached. **(E)** Cocontraction initiates in the braking phase (later than in the conditions with passive parallel stiffness; Fig 6), and achieves higher sustained levels post stabilisation with slower deactivation.

PLOS Computational Biology

**Table 3. Results of linear model fitting, including individual and interaction terms. Each term in the model is normalized to its range (Table 2). Coefficient estimates represent the relative "importance" of that term. Estimates and standard error are shown to one significant digit. Bolded values represent a coefficient estimate with magnitude > $10^{-1}$: *i.e.,* effects that are "important to first order". Note that all individual coefficients are important to first order except for activation and stiffness; however, together these have the only interaction coefficient with magnitude > $10^{-1}$.**

| Corresponding Term | Coefficient estimate | | |
| --- | --- | --- | --- |
| | All Terms | First-order terms only | First-order independent terms only |
| Intercept | **5e-1** | **5e-1** | **5e-1** |
| $\alpha_d$ | 2e-2 | | |
| $c_1$ | 6e-3 | | |
| $\alpha_a$ | **3e-1** | **3e-1** | **3e-1** |
| $v_{max}$ | **-2e-1** | **-2e-1** | **-2e-1** |
| $F_{max}$ | **-2e-1** | **-3e-1** | **-3e-1** |
| $\alpha_d : c_1$ | **1e-1** | 8e-2 | |
| $\alpha_d : \alpha_a$ | 3e-2 | | |
| $\alpha_d : v_{max}$ | -3e-2 | | |
| $\alpha_d : F_{max}$ | -4e-2 | | |
| $c_1 : \alpha_a$ | -2e-2 | | |
| $c_1 : v_{max}$ | -6e-3 | | |
| $c_1 : F_{max}$ | -6e-2 | | |
| $\alpha_a : v_{max}$ | 4e-2 | | |
| $\alpha_a : F_{max}$ | -6e-2 | | |
| $v_{max} : F_{max}$ | 5e-5 | | |
| $R^2$ | 0.95 | 0.94 | 0.92 |

associated with ageing– tend to decrease performance as well (although the exact effect depends on their interaction). To our knowledge, this is the first study to show that each of these five age-related changes to muscle can deleteriously affect performance.

Because of the model's relative simplicity, we can interrogate the solutions to understand the mechanistic links between parameter change and performance. In the following, we propose mechanistic reasons behind key findings: why activation time, peak force and maximum strain rate are the main contributors to performance; why deactivation time and stiffness interact; and why cocontraction, perhaps surprisingly, is associated with higher speed and overall performance in this model.

### Activation time, maximum force, and maximum muscle velocity are the main contributors to performance

Fig 4 shows that increases in activation time, decreases in peak force or decreases in maximum strain rate lead to reductions in performance (when holding other parameters constant). A linear fit of these three variables alone (without interactions but including the intercept; Table 3) explains 92% of the variation in performance, encapsulating their dominance in determining performance. The intercept value remains in place because the displacement error will remain at a constant value if the arm cannot move. The remaining 12 terms explain only an additional 3% of the variation in performance in the linear model. Our results suggest that age-related reductions in activation rate, maximum muscle strain rate, and isometric force all contribute overwhelmingly to reductions in ballistic performance. Because of the independence of these effects, we infer that a detectable reduction in any of these parameters with age generally has a negative effect on rapid point-to-point movement.

As the performance metric of RMSE is driven largely by time to target (Discussion and S2 Fig), these results come readily by considering each term's contribution to the time-averaged applied force. Since the load and initial distance to

target are constant, maximising time-averaged propulsive force is equivalent to minimising time. A lower activation time constant increases average force by allowing the muscle to reach full activation more quickly. A higher $F_{max}$ increases the force available from full activation; and a higher $v_{max}$ means the applied force is attenuated less as velocity increases.

In contrast, deactivation time and stiffness do not, on their own, affect performance in this dataset. They do, however, interact to affect performance, and only with each other (to first order; Table 3). These results may seem at first counterintuitive. Since the arm must slow down to reach the target, it may seem like slow deactivation would be detrimental– analogous to trying to stop a car with the accelerator pressed down. Moreover, since passive stiffness increases the applied force of the antagonist during acceleration and the agonist during braking, it may seem like increased passive stiffness should always be beneficial. Both these counterintuitive results can be explained by the force-velocity properties of muscle, as we discuss in the following sections.

### In the absence of passive stiffness, deactivation rate does not affect ballistic performance

If the agonist cannot turn off quickly during the braking phase, then it supplies excess force as the joint approaches its target. In order to stop in time, the controller must use one or a combination of three strategies: (1) the agonist reduces approach speed by reducing its applied force, (2) the antagonist applies braking force sooner, and/or (3) the antagonist applies extra braking force. The first strategy occurs in our simulations, but the effect is minor. As deactivation time increases, the agonist begins switching off only slightly earlier (by at most 15 ms; Fig 7B). The second strategy also occurs, but is likewise minor; as deactivation time increases, the antagonist comes online slightly earlier, but by at most 5 ms (Fig 7C). Despite these changes to force timing, the peak approach speed remains virtually unchanged (Fig 7B), and these small changes do not account for the similarity in performance.

The third strategy is much more effective– and is enabled by the muscle's force-velocity properties. Because both muscles have identical properties, and the antagonist operates on the eccentric part of the force-velocity curve (Fig 1A, 1C), it can always produce much larger force than the agonist as the joint is slowing down. Therefore, if the agonist remains stuck "on" due to slow deactivation, the antagonist can simply maintain heightened activation to overpower the agonist force (Fig 7D), leading to higher cocontraction (Fig 7E). The higher cocontraction effectively increases active damping at the joint [21,55]: if the antagonistic pair coactivates, a higher velocity is met with a higher force resisting motion, and this resistive force increases with the level of coactivation. This coactivation-dependent active damping compensates for the sustained propulsive force at large deactivation times. Therefore, slower deactivation rates lead to higher levels of cocontraction but not to lower performance– at least in the absence of stiffness.

### Stiffness and deactivation interact to enhance or reduce performance due to antagonist release

Increasing agonist stiffness can enhance performance, but only if the antagonist deactivation time is sufficiently small (Fig 5). This occurs because the initial flexed posture of the joint requires the agonist to be stretched, supplying passive torque about the joint (Fig 6A). To maintain stasis, the passive torque must be matched by antagonist active force (Fig 6B), acting as a muscular latch [56]. Movement initiation from the agonist stretches the antagonist, shifting it to the eccentric part of the force-velocity curve. If the deactivation time is low enough, the antagonist force can drop below the agonist passive force, and the net contribution to joint torque is positive (Fig 6C, cooler lines). If instead the deactivation time is high, the antagonist force quickly exceeds the passive contribution– there is then a net decrease in torque about the joint (Fig 6D warmer lines). Therefore, increasing stiffness can have a net positive or negative effect on performance, depending on the deactivation rate (Fig 5). If deactivation is slow, the antagonist slows the joint down, whereas if deactivation is high, the increased agonist stiffness provides extra positive torque about the joint, in excess of the antagonist's active force.

More specifically, our results suggest a critical deactivation time (approximately 45 ms; Fig 5) where increased stiffness has no effect on performance. In our model, if deactivation time is greater than this value, then age-related increases

in stiffness have a detrimental effect on performance; whereas if deactivation time is smaller, then increased stiffness enhances performance. This sensitivity to deactivation time has implications for dynamic simulation studies of musculoskeletal ageing. For example, the default OpenSim [57] deactivation time constant is 40 ms, used by Karami *et al.* [53] in a model of walking in the elderly. Using this value within our model would result in a slight positive effect of increased stiffness. Nowakowski *et al.* [54] increased $\alpha_d$ to 48 ms in their aged condition to assess fall risk in an OpenSim model. This value would indicate little to no effect of increased stiffness in our model. Thelen [40] and Murtola and Richards [38] used deactivation times of 60 and 65 ms respectively in their simulations of musculoskeletal ageing, with the latter finding a decrease in performance with increased stiffness, in accordance with Fig 5. Because of the sensitive interaction between passive stiffness and deactivation time, care should be taken in the selection of both parameters when inferring age-dependent effects.

Both increased deactivation rate and increased stiffness are associated with higher levels of coactivation and cocontraction. However, in our dataset cocontraction is not a feature to be avoided, but rather a behavioural outcome of maximising rapid movement performance.

## Cocontraction can be an optimal strategy for rapid movement

Arm reaching in elderly individuals is associated with higher levels of cocontraction [21,58,59], and is generally slower, both in reaction time and total movement time [19,60,61]. Ageing is also associated with reductions in neural conduction velocity [62,63], which may lead to feedback delays. Since cocontraction increases joint stability against perturbations without requiring feedback control [55,64], it has been suggested that elderly persons cocontract to stabilize their joints against perturbations [21], but that coactivation comes at a price of slowing movements [59,60,65,66] and reducing power production in older adults [61].

The present results offers another view. The current model has no outside perturbation, and no control deficits- the controller has an exact representation of the current states and system dynamics, and can change excitation to any value at 5 ms temporal resolution. And yet, cocontraction emerges naturally as an optimal strategy to maximize performance– which in this case is driven largely by minimizing time to target (S2 Fig). Indeed, trials with higher levels of coactivation tended to reach the target and stabilise *more quickly* than trials with lower levels of coactivation (S3 Fig).

In the idealised system under investigation, the controller provides an optimal sequence of muscle excitations that result in certain amounts of coactivation. By definition, any increase or decrease in coactivation level from this optimum would decrease performance. Our results suggest that, for a given musculoskeletal system, there exist optimal levels of coactivation and cocontraction for rapid point-to-point movements, which depend on the physiological and physical properties of the system. This implies that differences in coactivation between two biological systems (*e.g.,* young and old participants) do not indicate *a priori* whether either system is operating optimally for that task– the optimal level of coactivation depends on physics, physiology and the objective of the movement.

Our data suggest several heuristics for how and when coactivation is associated with performance differences. During the propulsive phase, the association between cocontraction and performance depends on the interaction between agonist stiffness and antagonist deactivation rate. Cocontraction will occur when the antagonist initially resists the agonist passive force. For a given level of stiffness, increased cocontraction is correlated with a loss of performance, as it indicates that the antagonist cannot deactivate quickly enough (Fig 6); but for two systems with different stiffnesses (with all other parameters similar), increased cocontraction may be associated with an increase or decrease in performance, depending on the muscle deactivation rate.

During the braking phase, increased deactivation time leads to increased cocontraction (Fig 7E). If all other parameters are equal, however, this does not affect performance, because cocontraction increases joint damping that compensates for relatively increased agonist activity. After stabilisation, optimal solutions with higher agonist activation during propulsion generally remain at higher levels of coactivation (Fig 3). In this dataset, because higher-performance trials tended to have

shorter ballistic phases and longer periods after braking, higher-performance tends to be associated with higher levels of coactivation (S3 Fig). Empirical evidence aligns with this finding; Suzuki *et al.* [50] found that faster point-to-point arm movements exhibited higher levels of post-movement coactivation.

In human and animal ballistic data, it is important to carefully consider how the absence of cocontraction would affect the performance dynamically, before claiming a causal link between the two. We have shown that, even in a simple, two-muscle model, the effects of cocontraction on movement time can be subtle and not immediately intuitive. The emergence of coactivation and cocontraction in this model, without any feedback delays or perturbations, suggest that increased coactivation in ageing may arise in part from increases in muscle stiffness and deactivation time, and that coactivation is not, in of itself, responsible for reductions in movement speed in the elderly.

This does not, however, preclude cocontraction as a compensatory strategy to reject perturbations without feedback; both a loss of rapid feedback control and the above physiological changes to muscle might lead to increased cocontraction in the elderly. The underlying causes of increased coactivation with age, and its associations with movement performance, will be clarified with further experiments, as well as with simulations that incorporate perturbation, signal noise or signal delays into the model.

### Limitations and future directions

The use of a simplified computational framework allowed us to explore a large space of muscle property changes that are associated with age. We were uniquely able to determine mechanistic links between parameters and performance because all aspects of the simulation are tightly controlled. Such analysis would be much more difficult, if not impossible, *in vivo* because ageing is a multifactorial condition wherein not all relevant parameters can be accurately measured, inferred or known.

The choice of a simplified model means that some fidelity is lost compared to biological data. The clearest example of this is in the predicted excitation patterns, which are much sharper than empirical EMG (Fig 2). Simulated excitation takes on this sharp, square-wave form because the cost function favours immediate maximum excitation to rapidly accelerate the arm towards the target, and maximally slow it close to the target (a "bang-bang" control strategy [48]). Empirical EMG instead shows a relatively slow rise and descent. EMG measures action potential propagation through the muscle, which is subject to dynamics not accounted for in step-wise excitations. In addition, reaching strategies show adaptations to energy-minimisation, even when rapid movement is rewarded [19], which are driven by energetic costs related to force-rate and work expenditure [52]. Augmenting the cost function with sum-squared-excitation (a crude approximation of a force-rate penalty) and work terms results in smoother excitation profiles (S4 Fig). While we opted to keep the cost function simple in the absence of a clear unbiased weighting between effort, speed and accuracy, future studies could expand this model to explore how neuromuscular changes affect effort-minimising reaching strategies with age.

While excitation and movement onset were coincident in the simulation, empirical EMG began rising about 50 ms before movement onset. This phenomenon, known as "electromechanical delay" (EMD) [67], arises from electrochemical and mechanical processes that show large inter-individual variation. One key contributor to EMD is tendon dynamics, which have not been modelled here. While the effect of ageing on tendon mechanics is complex and sometimes unclear [68], future studies could expand the performance landscape by incorporating tendon stiffness and viscoelastic effects.

Our study used a Hill-type model because such models are well-studied, widely used in reaching simulations [38,69,70] and are computationally cheap, making them well-suited for large parameter sweeps such as in the current study. However, more sophisticated muscle models [71,72] could be used in future work to address transient and hysteresis effects. Future models could also change task parameters by introducing perturbations or including energetic cost in the objective [19,52]. Further complicating features such as joint damping, variable moment arms, or differences between muscles could be implemented in follow-on studies, at the expense of computational cost.

 

Altogether, the simplifications inherent in the modelling approach, and the deviations from empirical behaviour (particularly in EMG patterns) suggest caution in extrapolating quantitative findings to human data. However, the broad patterns (*e.g.,* triphasic excitation, smooth reaching trajectories in realistic time) suggest that the model captures some qualitative features well. The simplicity of the model allows us to point to mechanistic links between physiological parameters and performance outcomes. These should be interpreted within the simplifying assumptions of the model itself, and their application to human and animal movement awaits confirmation from biological experimentation.

The influence of parameters on performance in the simulated landscape was explored using linear regression based on the standardized beta approach [46]. Because predictors are first normalised to their range in the linear regression, the magnitude of their regression coefficients (the "importance") could change if different parameter variation ranges were used in the analysis. For example, if $F_{max}$ were kept at relatively narrow values, it would appear to have less of an overall effect than $v_{max}$ and $\alpha_a$, if the latter terms maintained larger variation. Interaction effects could also change; if stiffness values were held at a narrow range of large values, then variation of deactivation time on its own would appear to have an effect on performance (Fig 5). The variation in muscle parameters considered in this simulation study is likely larger than natural variation associated with age in humans. For example, Runnels *et al.* [73] found that maximal isotonic knee extensor torque in young men is at most twice the value compared to their oldest cohort; whereas we explore a 3-fold change in maximum force. Thus, the absolute changes in performance in this simulation study should not be taken as representative of the changes expected with age– especially when considering the anatomical simplifications used in our model. The qualitative predictions are more likely to be generalisable, and suggest that changes to activation rate, isometric force and maximum strain rate detrimentally influence maximal performance with increasing age.

The current findings on the interaction between muscle parallel stiffness and deactivation rate warrant future experimental investigation. Reported age-related changes to muscle parallel stiffness can differ between studies and exhibit length-dependent effects [17,74]. Age-related changes to muscle deactivation rate are poorly studied, and the effects can differ between studies– even in the same isolated muscle preparations (*e.g.,* [5,10,13]). Because of this, and the interactivity of stiffness and deactivation rate, future experiments would be required to predict how they affect performance with age *in vivo*. In our dataset, simultaneous increases in stiffness and deactivation time can decrease or increase performance, depending on their correlation and location in the performance space (Fig 5). Despite the above uncertainty, our analysis shows both effects are hypothetically possible, and provides a mechanism why.

## Conclusions

We used a single-joint, two-muscle model to simulate point-to-point arm movements. Optimising excitations to minimise cumulative squared error to target resulted in ballistic motion with realistic triphasic activation. By altering muscle parameters that covary with age, we determined how parameter variation relates to changes in reaching performance. Reductions in force and maximum strain rate, and increases in activation time resulted in decreased performance, and these effects were largely independent of one another. In contrast, increasing parallel stiffness and deactivation time had little effect on performance on their own. For fast deactivation rates, increasing stiffness increases performance, whereas for slow deactivation rates increasing stiffness decreased performance. Both increased stiffness and deactivation times increased coactivation, but increased coactivation was generally associated with better performance in this dataset. This suggests that coactivation may not, in itself, limit movement speed with age, but may reflect increases in characteristic muscle deactivation time and stiffness– factors that can impair performance. Future studies will more precisely characterise age-related changes in muscle deactivation across species and explore how aged muscle affects perturbation response. In doing so, we move closer to uncovering the fundamental links between muscle physiology, motor control and locomotor performance through the lifespan.

## Supporting information

**S1 Fig. The fractional contribution of the ballistic phase to the overall performance metric ($J_{RMSE,rel}$) clusters strongly towards 1.** 90% of simulations have $J_{RMSE,rel} > 0.9999$, and none are below 0.99.
(EPS)

**S2 Fig. The performance metric of root-mean-square-error (RMSE) is positively correlated with both Time to Target ($T_r$) and Time to Stable Point ($T_s$).** Note that all conditions are able to reach either temporal condition within the simulation time window of 0.4 s.
(EPS)

**S3 Fig. Time to target, time to stable point, and RMSE against mean coactivation for all optimal simulations in the large parameter sweep (3920 conditions).** Higher coactivation is associated with lower time to target, lower time to stable point, and lower overall error. By any of these metrics, simulations with higher performance tend to be associated with higher levels of coactivation overall.
(EPS)

**S4 Fig. Optimal trajectory for an alternate cost function, compared to empirical data.** This figure is identical to Fig 2 except for the addition of a new simulation that augments the original cost function (Eq 3, for the baseline simulation in red) with terms for work and excitation squared ($J_{aug} = J + W_{exc}J_{exc} + W_{work}J_{work}$), where $W_{exc} = 10^{-5}$ m and $W_{work} = 10^{-4}$ m J$^{-1}$, for the simulation in blue), inspired by effort terms of force rate and work found to predict smooth arm movements [52]. Compared to the baseline simulation, these additional terms better match the empirical EMG data in a smoother excitation profile, and lower second agonist excitation peak. However, the post-movement excitation and duration of the antagonist burst is a worse fit to empirical EMG. Neither simulation matches the observed "electromechanical delay" [67] from EMG onset to elbow movement onset, likely because tendons and fiber action potentials are not modelled. $J_{exc} := \sum_1^{N-1} u_i^2$, and ($J_{work} := \Delta t \sum_1^N \text{smab}(\dot{l}_{ant,i}F_{ant,i}|s_w) + \text{smab}(\dot{l}_{ag,i}F_{ag,i}|s_w))$, where the timestep $\Delta t = 5$ ms, and the smooth absolute value function $\text{smab}(x|s) := x\tanh(xs)$, with $s_w = 100$ a smoothing parameter.
(EPS)

**S1 Appendix. Additional methodological details and analysis.** Included are mathematical details of force-length, force-velocity and activation characteristics, as well as analysis of the correlations between RMSE, time to target and coactivation.
(PDF)

## Acknowledgments

We thank Tiina Murtola for insightful feedback as this project developed, and Jeremy Wong for reviewing an earlier version of the manuscript.

## Author contributions

**Conceptualization:** Delyle T. Polet, Christopher T. Richards.

**Data curation:** Delyle T. Polet.

**Formal analysis:** Delyle T. Polet.

**Funding acquisition:** Christopher T. Richards.

**Investigation:** Delyle T. Polet.

**Methodology:** Delyle T. Polet.

**Project administration:** Delyle T. Polet.

**Resources:** Christopher T. Richards.

**Software:** Delyle T. Polet.

**Supervision:** Christopher T. Richards.

**Validation:** Delyle T. Polet.

**Visualization:** Delyle T. Polet.

**Writing – original draft:** Delyle T. Polet.

**Writing – review & editing:** Delyle T. Polet, Christopher T. Richards.

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
