## [Decision Letter · Decision Letter 0]

22 Oct 2025

PCOMPBIOL-D-25-01815

How muscle ageing affects rapid goal-directed movement: mechanistic insights from a simple model

PLOS Computational Biology

Dear Dr. Polet,

Thank you for submitting your manuscript to PLOS Computational Biology. After careful consideration, we feel that it has merit but does not fully meet PLOS Computational Biology's publication criteria as it currently stands. Therefore, we invite you to submit a revised version of the manuscript that addresses the points raised during the review process.

Please submit your revised manuscript within 60 days Dec 22 2025 11:59PM. If you will need more time than this to complete your revisions, please reply to this message or contact the journal office at ploscompbiol@plos.org. Please include the following items when submitting your revised manuscript:

We look forward to receiving your revised manuscript.

Kind regards,

Jian Liu

Academic Editor

PLOS Computational Biology

Andrea E. Martin

Section Editor

PLOS Computational Biology

**Additional Editor Comments:**

This is a promising contribution to the field. The manuscript would be significantly improved by providing further details on the model's implementation and validation, and more fully situating the work within the broader context of previous research

**Journal Requirements:**

Potential Copyright Issues:

i) Figure 1A. Please confirm whether you drew the images / clip-art within the figure panels by hand. If you did not draw the images, please provide (a) a link to the source of the images or icons and their license / terms of use; or (b) written permission from the copyright holder to publish the images or icons under our CC BY 4.0 license. Alternatively, you may replace the images with open source alternatives. See these open source resources you may use to replace images / clip-art:

**Reviewers' comments:**

Reviewer's Responses to Questions

**Comments to the Authors:**

Reviewer #1: In this paper the authors simplified the model described in Murtola and Richards (2022, 2023) to only include movements over the elbow joint. This single joint model was used to study a variety of parameters related to muscle activity and aging. The parameters that were examined and altered during the experiment were peak active force, maximum strain rate, activation time, deactivation time and stiffness. The parameter settings were selected based on previous knowledge and a thorough literature sweep. The simplified model applied was selected to enable us to explore the effects of varying parameters in a well-defined control task without having to analyze actual human data, which is noisy. They found four main terms (The Intercept term, Activation time, Maximum strain rate and Peak isometric force) and one interaction term (deactivation x stiffness) to be significant. Showing how a simplified model can be used to study specific parameters with settings representing different “ages”. In addition, they show that co-contractions might not only be a result of perturbations in older people but a result of optimizing performance when movement is constrained the way it is in older people.

Data code and availability: Checked the link at it works, it does.

Abstract: I am missing the information that the model was created to represent movement over the elbow joint. Please include this information in the abstract.

Introduction: I like the introduction and think that you have structured it nicely.

Method:

• Line 67-68: Please specify which joint you are working with, in order to make it easier to follow. You could just put it in parenthesis, like this: ….rotational joint (the elbow joint) from one…

• Line 139-159: The method selected here implies that you assume the relationship between the parameters to be independent and linear. I would be more confident about the results if the data was fitted to a non-linear multivariate regression model as well, since this would allow for more complex relationships. When working with purely isometric contractions the linear models have been enough to describe e.g., Force and EMG activity, but when moving into more naturalistic movement non-linear models have become required to describe the data. When looking at the supplementary correlation plots there also seems to be a simplification to call some of these relationships linear.

• You apply a method where you change one parameter at a time. This has been shown to mask true performance optimum. In addition, it fails to account for interactions between parameters, which you clearly state that you assume that there isn’t. However, another way to solve this would be by applying design of experiment and altering the parameters in a designed fashion. Even in a simple system I would assume the studied factors to be interdependent and have a complex correlation pattern. I suggest that you apply a different linear regression method that allow for the factors to be studied simultaneously.

• Model significance is not specified. You have not specified how you evaluate the quality of the models. Please add aa segment to the method describing how you made sure that the models were significant.

Results:

• Figure 2: Were the empirical data preloaded in any way? Were the simulations loaded in a similar fashion? Does this simulation represent the baseline conditions?

• Figure 5: RMSE varies between 0.07 and 0.0735. How significant is this difference? Interpreting this as “increasing stiffness decreases performance” sounds like a stretch, these are marginal differences highlighted by changes in color. Please provide the rational for how such a small increase in RMSE is indicative of “increasing stiffness decreases performance” (line 199). Even when taking into account that you have standardized your data to have a standard deviation =1, this seems small.

• How do you evaluate the significance of the main model where you obtain Table 3?

• Line 235-242 You mention R2, but not Q2? It is indeed important to have a well-fitted model, but also to have it explain how well your model predicts data. Why is not Q2 reported? Why is R2 (and Q2) not reported for your main model?

Discussion:

• Nicely structured and clear. I only miss one thing. You should include discussion about the timing of the empirical compared to simulations, where the simulation activity occurred much slower compared to the empirical (Figure 2). What is the impact of this on your results?

Reviewer #2: This manuscript investigated the effect of parameters for muscle contraction on a single joint reaching movement via a computer simulation. There results show that there is a non-trivial interaction between the stiffness of parallel elastic component and muscle deactivation rate, which is very interesting, However, I have some suggestions.

Introduction

The authors use a simplified reaching as a model movement. But, in Introduction, they discuss performance of locomotion and fall risk, which is more relevant to lower extremity. That seems to be a big stretch to me since mechanical demand imposed on arms during reaching is completely different from that on leg muscles during walking or running. The authors could focus on the argument mainly with reaching movement, which would be more natural to me.

Methods

Their model does not have a series elastic component (SEC). Due to SEC, the phase shift between the entire muscle length and fascicles can be significant in lower extremity during fast calf-raise movement (e.g. Takeshita et al. 2006), but maybe not so significant in forearm movement (Amis et al. 1987). While the justification seems to be discussed in the previous study, the authors should also provide their own justification.

References

J Appl Physiol. 2006 Jul;101(1):111-8. doi: 10.1152/japplphysiol.01084.2005. Epub 2006 Feb 23. Resonance in the human medial gastrocnemius muscle during cyclic ankle bending exercise. Daisuke Takeshita 1, Akira Shibayama, Tetsuro Muraoka, Tadashi Muramatsu, Akinori Nagano, Tetsuo Fukunaga, Senshi Fukashiro

J Physiol 1987 Aug:389:37-44. doi: 10.1113/jphysiol.1987.sp016645. Relative displacements in muscle and tendon during human arm movements. A Amis 1, A Prochazka, D Short, P S Trend, A Ward

The authors should provide the properties of input signals. Is the value of input signals at each time bin determined or some parameterizations were used etc.

Minor comments:

L 118: “(see Supplemental Information)” should refer to a particular section of the supplemental info.

**Have the authors made all data and (if applicable) computational code underlying the findings in their manuscript fully available?**

Reviewer #1: Yes

Reviewer #2: Yes

PLOS authors have the option to publish the peer review history of their article (what does this mean? ). If published, this will include your full peer review and any attached files.

**Do you want your identity to be public for this peer review?** For information about this choice, including consent withdrawal, please see our Privacy Policy .

Reviewer #1: No

Reviewer #2: **Yes:** Daisuke Takeshita

**Figure resubmission:**

**Reproducibility:**



---

## [Decision Letter · Decision Letter 1]

15 Feb 2026

Dear Dr Polet,

We are pleased to inform you that your manuscript 'How muscle ageing affects rapid goal-directed movement: mechanistic insights from a simple model' has been provisionally accepted for publication in PLOS Computational Biology.

Best regards,

Jian Liu

Academic Editor

PLOS Computational Biology

Andrea E. Martin

Section Editor

PLOS Computational Biology

Reviewer's Responses to Questions

**Comments to the Authors:**

Reviewer #1: I am satisfied with the changes that the authors have made to the paper. The methodology is now clear to me. I appreciate the additions that have been made to the manuscript and have received adequate answers to the questions I raised.

Reviewer #2: The authors have properly addressed my suggestions. So, no further comments from me.

**Have the authors made all data and (if applicable) computational code underlying the findings in their manuscript fully available?**

Reviewer #1: Yes

Reviewer #2: Yes

PLOS authors have the option to publish the peer review history of their article (what does this mean? ). If published, this will include your full peer review and any attached files.

**Do you want your identity to be public for this peer review?** For information about this choice, including consent withdrawal, please see our Privacy Policy .

Reviewer #1: No

Reviewer #2: **Yes:** Daisuke Takeshita

---

## [Editor Report · Acceptance letter]

PCOMPBIOL-D-25-01815R1

How muscle ageing affects rapid goal-directed movement: mechanistic insights from a simple model

Dear Dr Polet,

I am pleased to inform you that your manuscript has been formally accepted for publication in PLOS Computational Biology. Your manuscript is now with our production department and you will be notified of the publication date in due course.

With kind regards,

Zsofia Freund
